# Exploiting Space Folding by Neural Networks

## Abstract

Recent findings suggest that consecutive layers of neural networks with the ReLU activation function *fold* the input space during the learning process. While many works hint at this phenomenon, an approach to quantify the folding was only recently proposed by means of a space folding measure based on the Hamming distance in the ReLU activation space. Moreover, it has been observed that space folding values increase with network depth when the generalization error is low, but decrease when the error increases, thus underpinning that learned symmetries in the data manifold (visible in terms of space folds) contribute to the network's generalization capacity. Inspired by these findings, we propose a novel regularization scheme that enforces folding early during the training process. Further, we generalize the space folding measure to a wider class of activation functions through the introduction of equivalence classes of input data. We then analyze its mathematical and computational properties and propose an efficient sampling strategy for its implementation. Lastly, we outline the connection between learning with increased folding and contrastive learning, hinting that the former is a generalization of the latter. We underpin our claims with an experimental evaluation.

## 1 Introduction

Biological sensory systems, as well as artificial neural networks, transform the input signal into internal representations that efficiently and effectively capture information needed for current and future tasks. For example, in the human eye, the retina removes redundant spatiotemporal structure from incoming light so that it may be efficiently transmitted through the optic nerve. This representation is then transformed in the cortical area by extracting frequently occurring features in support of efficient coding and discrimination of natural images (Barlow, 1961; Atick & Redlich, 1990; van Hateren, 1992; Meister et al., 1995; Balasubramanian & Berry, 2002; Puchalla et al., 2005; Doi et al., 2012). Similarly to biological structures, artificial neural networks also transform its input signal, allowing for its use for downstream tasks. This transformation can be analytically studied with tools developed for signal processing, see e.g., Mallat (1989; 2009; 2012).

Motivated by this, we aim to study artificial neural networks (ANNs) through the lens of how they transform the input space. Recent works indicate that ANNs *fold* the input space during the training process, meaning that distant input samples can become close in the activation space (Montúfar et al., 2014; Keup & Helias, 2022). Building on these ideas, Lewandowski et al. (2025) proposed a range-based measure in the discrete activation space of ReLU neural networks to quantify *how much* a network folds its input space as it learns. Their analysis focuses on deviations from convexity when mapping a straight-line path in the Euclidean input space to the Hamming activation space. Though simple in nature, analyzing paths in the activation space prove to be insightful as it can be applied to arbitrary paths, and thus statistics derived from these probes might capture the global nature of folds while remaining computationally tractable (Freeman & Bruna, 2016). Hence, what appears to be a restrictive slice through the input space, instead functions as an a stethoscope of sorts that lets us listen to how (and how much) the network convolutes the input space during learning.

In Lewandowski et al. (2025), the authors rely on the intuition that, if the data manifold learned by a neural network is flat, then the (Euclidean) distance increases monotonically with respect to the initial point when walking along a straight line connecting these points. Contrary, when the data manifold is folded, then

the (Hamming) distance computed between respective activation patterns in the Hamming activation space (defined in Sec. 3) changes non-monotonically – at some point the network "refolds" such that the Hamming distance decreases – indicating that two previously distant (in the input space) data points have come closer (cf. Fig. 1 right and Fig. 2). Originally developed for ReLU networks, this approach leverages the fact that the ReLU activation function partitions the input space into disjoint linear regions (Makhoul et al., 1989; Montúfar et al., 2014). In our paper, we firstly show that these regions correspond to equivalence classes defined by the pre-images of either $\{0\}$ or the strictly positive interval $(0, \infty)$. Extending $\{0\}$ to $(-\infty, 0]$ provides a straightforward generalization to a broader class of monotonic activation functions. Secondly, we derive several properties of the space folding measure $\chi$, which do also hold in the general case. Thirdly, since computing $\chi$ relies on sampling from different activation regions, we introduce a non-parametric sampling algorithm that exploits the structure of the aforementioned equivalence classes, thereby reducing redundant computations. Lastly, we leverage the fact that space folding values have been observed to *increase* with network depth when the generalization error is low, but *decrease* when the error increases (Lewandowski et al., 2025). We thus hypothesize that an increased folding enhances the network's generalization capabilities, and introduce a novel regularization strategy that applies the folding measure at periodic intervals during training (e.g., every $n$ training epochs) to induce stronger folding in the early stages and diminish its influence later in training. Our contributions are as follows.

- **Generalized Folding Measure:** We generalize the space folding measure beyond the ReLU activation function. Our approach relies on the fact that the pre-image of the partition $\{(-\infty, 0], (0, \infty)\}$ divides the domain into two connected sets, $f^{(-1)}((-\infty, 0])$ and $f^{(-1)}((0, \infty))$, for any monotonic and continuous $f$.

- **Theoretical Analysis:** We state and prove general properties of the folding measure, such as (*i*) its stability under traversing different activation regions, (*ii*) the sufficient and necessary, i.e., characterizing, condition for flatness, (*iii*) its sensitivity to the direction of the path, (*iv*) invariance of flatness to direction of path, (*v*) non-additivity.

- **New Regularization:** We introduce a new regularization procedure for training neural networks by penalizing low space folding values. We aim to exploit the previously observed link between folding and generalization and enforce folding during the training process. To this goal, we propose a differentiable approximation of the original folding measure.

The remainder of the paper is organized as follows. Sec. 2 outlines related work; Sec. 3 introduces necessary concepts and fixes notation for the remainder of the paper; Sec. 4 recalls the definition of the space folding measure and then provides detailed analysis paired with the introduction of the global folding measure Sec. 5 contains the performed experiments; Sec. 6 outlines an approach on how to use the folding measure for regularization during training; and finally, Sec. 7 we conclude our work and outline future research directions. In Appendices, Sec. A sketches sensitivity analysis of the folding measure to perturbations in activation patterns along a path; Sec. B introduces a sampling technique from activation paths along a 1D path which relies on the Hamming distances between samples; Sec. C describes an impact of batch normalization and dropout on the behaviour of the folding measure; lastly, Sec. D provides results of folding using ELU activation function.

## 2   Related Work

**Folding.**   The idea of folding the (input) space has been investigated, among others, in computational geometry (Demaine et al., 2000). In the context of neural networks, Montúfar et al. (2014) argued that each hidden layer in a ReLU neural network acts as a folding operator, recursively collapsing input-space regions. In Phuong & Lampert (2020), the authors defined the folds by ReLU networks, but left the exploration quite early on. Lewandowski et al. (2025) proposed the first measure to quantify the folding by ReLU neural networks, but it was restricted to ReLU networks and lacked a corresponding theoretical analysis. In our work, we generalize the measure beyond the ReLU activation function, and then exploit it for regularization.

We remark that folding can be seen as a process generating symmetry: When the neural network folds the input space, it effectively identifies different inputs (e.g., an image and its mirror) by mapping them to the same activation pattern – a form of learned invariance. Somewhat implicitly, symmetries have been at the core of some of the most successful deep neural network architectures, e.g., CNNs (Fukushima, 1980; LeCun et al., 1989) are equivariant to translation invariance characteristic of image classification tasks, while GNNs (Battaglia et al., 2018) are equivariant to the full group of permutations (see Higgins et al. (2022) for a detailed overview). Our work analyzes symmetries (reflection groups) that arise by space folding and their impact on the generalization capacity of the model.

**Path Analysis.** Fawzi et al. (2018) used path analysis between input data to explore whether there exists a continuous path that remains in the decision region between any two points of the same label. Hénaff et al. (2019) proposed that the visual system transforms inputs to follow straighter temporal trajectories, and developed a methodology for estimating the curvature of an internal trajectory from human perceptual judgments. In Hosseini & Fedorenko (2023), the authors developed a curvature metric that relies on the neural trajectory of words (tokens) in a sentence and found a quantitatively behavior of the metric in trained models. Goujon et al. (2024) showed that along one-dimensional paths, nonlinearity points scale linearly with depth, width, and activation complexity, while Gamba et al. (2022) proposed a direction-based method to recover all the linear regions along a path. Similarly to these works, we focus on path analysis and its descriptive statistics, however, in addition we leverage the underlying geometry of data. In this way, we capture the transformation of the space by neural networks.

## 3 Preliminaries

We define a *ReLU neural network* $\mathcal{N} : \mathcal{X} \to \mathcal{Y}$ with the total number of $N$ neurons as an alternating composition of the ReLU function $\sigma(x) := \max(x, 0)$ applied element-wise on the input $x$, and affine functions with weights $W_k$ and biases $b_k$ at layer $k$. An input $\mathbf{x} \in \mathcal{X}$ propagated through $\mathcal{N}$ generates non-negative activation values on each neuron. A *binarization* is a mapping $\pi : \mathbb{R}^N \to \{0, 1\}^N$ applied to a vector $v = (v_1, \ldots, v_N) \in \mathbb{R}^N$, resulting in a binary vector by clipping strictly positive entries of $v$ to 1, and non-positive entries to 0, that is $\pi(v_i) = 1$ if $v_i > 0$, and $\pi(v_i) = 0$ otherwise. In our case, the vector $v$ is the concatenation of all neurons of all hidden layers and its binarization, called an *activation pattern*, represents an element in a binary hypercube $\mathcal{H}^N := \{0, 1\}^N$ where the dimensionality is equal to the number $N$ of (hidden) neurons in network $\mathcal{N}$. A *linear region* is an element of a partition covering the input domain where the network behaves as an affine function (Fig. 1, left). The Hamming distance, $d_H(u, v) := |\{u_i \neq v_i \text{ for } i = 1, \ldots, N\}|$, measures the difference between $u, v \in \mathcal{H}^N$, and for binary vectors is equivalent to the $L_1$ norm between those vectors. Lastly, as we will deal with paths of activation patterns, we denote the operation of joining those paths with the operator $\oplus : \mathcal{H}^{k \cdot N} \times \mathcal{H}^{(n-k+1) \cdot N} \to \mathcal{H}^{n \cdot N}$ such that $\{\pi_1, \ldots, \pi_k\} \oplus \{\pi_k, \ldots, \pi_n\} = \{\pi_1, \ldots, \pi_k, \ldots, \pi_n\}$. The operation $\oplus$ is defined for connected paths, where the last activation pattern of one path matches the first activation pattern of the other.

## 4 Space Folding Measure: Construction and Properties

### 4.1 Construction

Consider a straight line connecting two input points $\mathbf{x}_1, \mathbf{x}_2$ in the Euclidean input space. The intermediate points are realized by varying the parameter $t$ in a convex combination $(1 - t)\mathbf{x}_1 + t\mathbf{x}_2$. For a practical implementation, Lewandowski et al. (2025) spaced the parameter $t$ equidistantly on $[0, 1]$, creating $n$ segments. Equal spacing, though easy and fast to implement, frequently results in suboptimal choice of the intermediate points (we address this issue in Appendix B). To obtain a walk through activation patterns, we map the straight line $[\mathbf{x}_1, \mathbf{x}_2]$ through a neural network $\mathcal{N}$ to a *path* $\Gamma := \{\pi_1, \ldots, \pi_n\} \in \mathcal{H}^{n \cdot N}$ in the Hamming activation space, where the intermediate activation patterns belong to a binary hypercube, $\pi_i \in \mathcal{H}^N$ for all $i \in \{1, \ldots, n\}$ (see Fig. 2). We consider a change in the Hamming distance with respect to the initial activation pattern $\pi_1$ at each step $i$, $\Delta_i := d_H(\pi_{i+1}, \pi_1) - d_H(\pi_i, \pi_1)$, and then look at the maximum of the

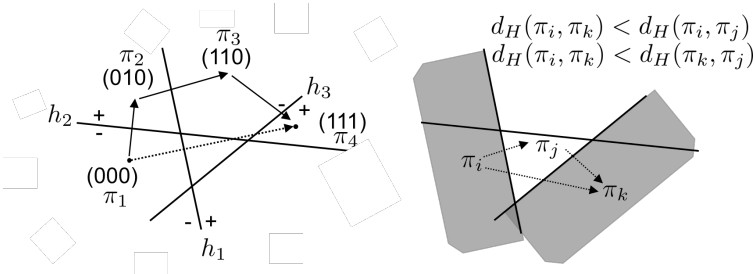

Figure 1: **Left:** Illustration of a walk on a straight path in the Euclidean input space and the Hamming activation space. The dotted line represents the shortest path in the Euclidean space. The arrows represent *a* shortest path in the Hamming distance between activation patterns $\pi_1$ and $\pi_4$ (in the Hamming space the shortest path is not unique). **Right:** Symmetry in the activation space: gray regions are closer to each other in the Hamming distance than to the region $\pi_j$ that lies between them.

cumulative change $\max_k \sum_{i=1}^k \Delta_i$ along the path $\Gamma$,

$$r_1(\Gamma) = \max_i \sum_{j=1}^i \Delta_j = \max_i d_H(\pi_i, \pi_1). \tag{1}$$

We further keep track of the total distance traveled on the hypercube when following the path,

$$r_2(\Gamma) = \sum_{i=1}^{n-1} d_H(\pi_i, \pi_{i+1}). \tag{2}$$

For a measure of *space flatness*, we consider the ratio $r_1(\Gamma)/r_2(\Gamma)$. Equivalently, the *space folding* measure results as

$$\chi(\Gamma) := 1 - \max_i d_H(\pi_i, \pi_1) / \sum_{i=1}^{n-1} d_H(\pi_i, \pi_{i+1}). \tag{3}$$

The folding measure is lower and upper bounded, $\chi \in [0, 1]$ Lewandowski et al. (2025). We will now formally define folding of the space, and the we will generalize the folding measure to any monotonic activation function.

**Definition 4.1** (**Space Folding**). We say that the input space is *folded* between inputs $\mathbf{x}_1$ and $\mathbf{x}_2$ with activation patterns $\pi_1$ and $\pi_2$, respectively, if $\chi(\Gamma) > 0$ for a path $\Gamma$ spanned between $\pi_1$ and $\pi_2$.

### 4.2 Beyond ReLU

Before stating several properties of the folding measure $\chi$, we interpret a walk through activation regions in ReLU-based MLP as a walk traversing distinct equivalence classes, and then show how this extends to *any* activation function. This makes our study directly applicable to vast range of activation functions. We start by defining the input equivalence relationship for ReLU neural networks. We will abuse the notation slightly by using $\pi(\mathbf{x})$ to denote an activation pattern of an input $\mathbf{x}$.

**Definition 4.2.** We say that two inputs $\mathbf{x}_1, \mathbf{x}_2$ are in an equivalence relationship with respect to a neural network $\mathcal{N}$ if their activation patterns $\pi_1, \pi_2$ are the same, i.e.,

$$\mathbf{x}_1 \sim_{\mathcal{N}} \mathbf{x}_2 \iff d_H(\pi_1, \pi_2) = 0.$$

For ReLU neural networks the equivalence class $[\mathbf{x}_1]_{\mathcal{N}} := \{\mathbf{z} \in \mathbb{R}^m \mid \mathbf{z} \sim_{\mathcal{N}} \mathbf{x}_1\}$ corresponds to a linear region which contains point $\mathbf{x}_1$. We now show that the relation in Def. 4.2 is that of equivalence. Indeed, *reflexivity* holds as $\mathbf{x} \sim \mathbf{x} \Rightarrow \pi(\mathbf{x}) = \pi(\mathbf{x}) \Rightarrow d_H(\pi(\mathbf{x}), \pi(\mathbf{x})) = 0$, and vice-versa, $d_H(\pi(\mathbf{z}), \pi(\mathbf{x})) = 0$ holds for all $\mathbf{z}$ such that $\mathbf{z} \in [\mathbf{x}]_{\mathcal{N}}$, which also contains $\mathbf{x}$. *Symmetry* is straightforward to check, and *transitivity* holds as

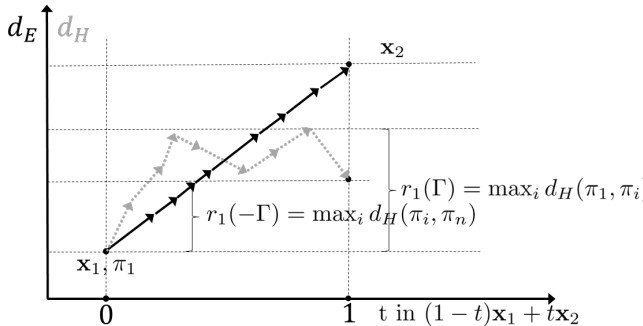

Figure 2: 1D straight walk from $\mathbf{x}_1$ to $\mathbf{x}_2$ in the Euclidean space (black full arrows) and the Hamming activation space (gray dotted arrows). Observe that in the Hamming activation space it might happen that $d_H(\pi_1, \pi_n) < \max_i d_H(\pi_1, \pi_i)$, which indicates space folding. The steps are optimized to visit each equivalence class exactly once (not equidistant).

$\mathbf{x} \sim \mathbf{y}$ and $\mathbf{y} \sim \mathbf{z}$ implies that $d_H(\pi(\mathbf{x}), \pi(\mathbf{y})) = 0$ and $d_H(\pi(\mathbf{y}), \pi(\mathbf{z})) = 0$, thus also $d_H(\pi(\mathbf{x}), \pi(\mathbf{z})) = 0$, and inversely, zero Hamming distances between $\pi(\mathbf{x})$ and $\pi(\mathbf{y})$ as well as $\pi(\mathbf{y})$ and $\pi(\mathbf{z})$ imply that $\mathbf{z} \in [\mathbf{x}]_\mathcal{N}$.

Definition 4.2 paves a way to extending the folding analysis to a richer class of activation functions. However, we lose the geometrical interpretation of equivalence classes as "linear regions". Henceforth for the computation of $\chi$, we consider a walk through input equivalence classes, not linear regions, thus extending the applicability of the space folding measure to much wider class of neural architectures. In order to obtain binary activation vectors, we clip the values on the hidden layers (after applying the activation function) in a similar way as with the ReLU function, i.e., for a vector of activation values $\mathbf{a} \in \mathbb{R}^n$ we create an *activation pattern* by only considering strictly positive vs. non-positive activation values, and denoting them with 1 and 0, respectively. We remark that Balestriero & Baraniuk (2018) extended a framework originally developed for studying ReLU neural networks by links to probabilistic Gaussian Mixture Models (GMMs) and Vector Quantization techniques (VQ). In that interpretation, piecewise affine, convex nonlinearities like ReLU, absolute value, and max-pooling can be interpreted as solutions to certain natural "hard" VQ inference problems, while sigmoid, hyperbolic tangent, and softmax can be interpreted as solutions to corresponding "soft" VQ inference problems. While this is an interesting idea, we believe that our approach is inherently simpler and more intuitive.

### 4.3 Properties

In this section, we look at several properties of the folding measure. They will serve as the base for further analysis.

**Lemma 4.3.** *The folding measure $\chi$ has the following properties:*

1. *(**Stability.**) Multiple steps in the same activation region do not influence $\chi$.*

2. *(**Flatness Condition.**) $\chi(\Gamma) = 0$ implies that $d_H(\pi_1, \pi_i)$ is increasing for $i = 1, \ldots, n$ along $\Gamma$.*

3. *(**Asymmetry.**) The folding measure is sensitive to the direction of traversal, i.e., $\chi(\Gamma) \neq \chi(-\Gamma)$.*

4. *(**Flatness Invariance.**) $\chi(\Gamma) = 0$ if and only if $\chi(-\Gamma) = 0$ for a path $\Gamma = \{\pi_1, \ldots, \pi_n\}$.*

5. *(**Non-additivity.**) The folding measure $\chi$ in general is neither sub-additive nor super-additive over concatenated path, i.e., it neither holds that $\chi(\Gamma_1 \oplus \Gamma_2) > \chi(\Gamma_1) + \chi(\Gamma_2)$ nor $\chi(\Gamma_1 \oplus \Gamma_2) < \chi(\Gamma_1) + \chi(\Gamma_2)$, where the operator $\oplus$ is as define in Sec. 3.*

*Proof.* We now prove properties listed in Lemma 4.3.

1. (**Stability.**) The proof is straightforward as staying in the same activation region does not change either Hamming distance.

2. (**Flatness Condition.**) Assume, without loss of generality, that each $\pi_i$ is distinct from $\pi_{i-1}$. By triangle inequality, $d_H(\pi_1, \pi_n) \leq d_H(\pi_1, \pi_i) + d_H(\pi_i, \pi_n)$. If $d_H(\pi_1, \pi_i) \leq d_H(\pi_1, \pi_{i-1})$ were to hold for any $i$, then by applying triangle inequality again, we would find that $d_H(\pi_1, \pi_n) < \sum_{i=1}^{n-1} d_H(\pi_i, \pi_{i+1})$. However, $\chi(\Gamma) = 0$ implies that $d_H(\pi_1, \pi_n) = \sum_{i=1}^{n-1} d_H(\pi_i, \pi_{i+1})$, leading to a contradiction.

3. (**Asymmetry.**) By counterexample: consider $\Gamma = \{\pi_1, \pi_2, \pi_3\}$, where $\pi_1 = (000), \pi_2 = (111), \pi_3 = (001)$, and its reverse $-\Gamma = \{\pi_3, \pi_2, \pi_1\}$. Then, $r_2(\Gamma) = r_2(-\Gamma)$ but $r_1(\Gamma) = 3$ and $r_1(-\Gamma) = 2$, thus $\chi(\Gamma) \neq \chi(-\Gamma)$.

4. (**Flatness Invariance.**) Observe that it is sufficient to prove it only in way direction as we can re-index the path $\Gamma$ to obtain its reverse. If $\chi(\Gamma) = 0$, then $d_H(\pi_1, \pi_i)$ never decreases with increasing $i$. Along the reversed path $-\Gamma$, this translates to $d_H(\pi_n, \pi_{n-i+1})$ never decreasing, so $\chi(-\Gamma) = 0$. Conversely, if $\chi(-\Gamma) = 0$, then similarly $\chi(\Gamma) = 0$.

5. (**Non-additivity.**) For a counter example of the sub-additivity consider paths $\Gamma_1 = \{\pi_1, \pi_2\}$ and $\Gamma_2 = \{\pi_2, \pi_3, \pi_4\}$ with the activation regions defined as

$$\pi_1 = \begin{pmatrix} 0 \\ 0 \\ 0 \end{pmatrix}, \pi_2 = \begin{pmatrix} 0 \\ 0 \\ 1 \end{pmatrix}, \pi_3 = \begin{pmatrix} 1 \\ 1 \\ 1 \end{pmatrix}, \pi_4 = \begin{pmatrix} 1 \\ 0 \\ 1 \end{pmatrix}. \tag{4}$$

In this case, $\chi(\Gamma_1 \oplus \Gamma_2) = \frac{1}{4}$ and $\chi(\Gamma_1) + \chi(\Gamma_2) = 0 + \frac{1}{3} = \frac{1}{3}$, thus $\chi(\Gamma_1) + \chi(\Gamma_2) \geq \chi(\Gamma_1 \oplus \Gamma_2)$ (for connected paths $\Gamma_1$ and $\Gamma_2$). To see that we can also construct a counter example for super-additivity, consider paths as previously with the activation patterns defined as

$$\pi_1 = \begin{pmatrix} 0 \\ 0 \\ 0 \end{pmatrix}, \pi_2 = \begin{pmatrix} 1 \\ 1 \\ 1 \end{pmatrix}, \pi_3 = \begin{pmatrix} 0 \\ 0 \\ 1 \end{pmatrix}, \pi_4 = \begin{pmatrix} 1 \\ 0 \\ 0 \end{pmatrix}, \tag{5}$$

Then, $\chi(\Gamma_1 \oplus \Gamma_2) = \frac{4}{7}$ while $\chi(\Gamma_1) + \chi(\Gamma_2) = 0 + \frac{1}{2} = \frac{1}{2}$, thus $\chi(\Gamma_1) + \chi(\Gamma_2) \leq \chi(\Gamma_1 \oplus \Gamma_2)$.

□

In the remainder of this section, we will discuss and interpret properties listed in Lemma 4.3.

**Stability:** The stability property justifies introducing an equivalence relationship between inputs $\mathbf{x}_1$ and $\mathbf{x}_2$ with no folding between, which we formalize as follows.

**Definition 4.4.** We say that input points $\mathbf{x}_1$ and $\mathbf{x}_2$ are equivalent under $\chi$ if

$$\mathbf{x}_1 \sim_\chi \mathbf{x}_2 \Leftrightarrow \chi(\Gamma(\mathbf{x}_1, \mathbf{x}_2)) = 0, \tag{6}$$

where $\Gamma(\mathbf{x}_1, \mathbf{x}_2)$ is a path of activation patterns spanned between $\mathbf{x}_1$ and $\mathbf{x}_2$.

We will use Def. 4.4 to introduce a space folding-based metric in the Hamming activation space in Sec. 4.4.

**Flatness Condition:** The Flatness condition stated in Lemma 4.3 implies that folding occurs if $r_1$ (Eq. (1)) decreases at least once along the path. Flatness means that a straight line mapped through a network is itself a "straight line" in the Hamming space.

**Non-additivity.** While neither super- nor sub-additivity holds for every path $\Gamma$, in our experiments we have only observed sub-additivity of the folding measure. The counterexample for super-additivity (Eq. (5)), seems to be a rare occurrence in trained networks, though it can be observed in specially constructed examples (see CantorNet by Lewandowski et al. (2024)). The general lack of super- or sub-additivity, but empirical sub-additivity motivates us to introduce the interaction coefficient $\mathcal{I}$ (deviation from additivity) for two paths $\Gamma_1$ and $\Gamma_2$ as $\mathcal{I} : \mathcal{H}^{n_1 \cdot N} \times \mathcal{H}^{n_2 \cdot N} \to [0, 1]$, where

$$\mathcal{I}(\Gamma_1, \Gamma_2) := |\chi(\Gamma_1 \oplus \Gamma_2) - \chi(\Gamma_1) - \chi(\Gamma_2)|. \tag{7}$$

$\mathcal{I}(\Gamma_1, \Gamma_2) = 0$ for paths $\Gamma_1, \Gamma_2$ if and only if additivity holds for those two subpaths; $\mathcal{I}(\Gamma_1, \Gamma_2) > 0$ means some fold "cancel out" or amplify when combining paths. In Sec. 5 we explore the applicability of the interaction coefficient $\mathcal{I}$ to study the geometry of the decision boundary. Our intuition is as follows: Folding of the input space is related to *deviations* from convexity (see Moser et al. (2022); Lewandowski et al. (2025)) and the increased number of these deviations indicates a decision boundary that is more ragged, and thus more sensitive to small perturbations of the original data. We hypothesize that $\mathcal{I}$ will exhibit sensitivity when computed between an original sample and its adversarial perturbation, and present preliminary experiments underpinning our hypothesis in Sec. 5.

While $\chi$ proves to be asymmetric in nature, in our experiments we have observed that, as the number of neurons increases, $\chi(\Gamma)$ and $\chi(-\Gamma)$ seem to converge. We hypothesize that with many neurons, any specific order of folds can be realized in reverse by alternate paths due to the abundance of intermediate regions. We express it as Conjecture 4.5.

**Conjecture 4.5.** *As neural network's total number of neurons $N \to \infty$, the folding measure becomes asymptotically symmetric, i.e., $|\chi(\Gamma) - \chi(-\Gamma)| \to 0$.*

We now introduce the notion of *sparsity* of folding values as the ratio of paths that exhibit no folding effects, to the paths along which the folding is positive.

**Definition 4.6** (**Sparsity**). Let $|A|$ denote the cardinality of a set $A$. The sparsity $\mathcal{S}_\mathcal{N}$ of $\chi$ under $\mathcal{N}$ is the ratio

$$\mathcal{S}_\mathcal{N} := \frac{|\{\Gamma : \chi(\Gamma) = 0\}|}{|\{\Gamma\}|} \in [0, 1]. \tag{8}$$

We empirically investigate the sparsity as a function of total number of neurons in Sec. 5.

### 4.4 Space Folding-based Pseudo-metric

In this section we introduce a pseudo-metric whose construction is inspired by $\chi$ (Eq. (3)). The path $\Gamma$ is spanned between its edge points, i.e., $\Gamma = \Gamma(\mathbf{x}_1, \mathbf{x}_2)$. Without loss of generality, we assume that every intermediate step visits exactly one activation pattern.

**Proposition 4.7** (Space Folding-based Pseudo-metric). *Let $d_\chi$ be a symmetrized space folding measure, i.e.,*

$$d_\chi(\mathbf{x}_1, \mathbf{x}_2) := \frac{\chi(\Gamma(\mathbf{x}_1, \mathbf{x}_2)) + \chi(\Gamma(\mathbf{x}_2, \mathbf{x}_1))}{2} = \frac{1}{2} \frac{\max_j d_H(\pi_1, \pi_i) + \max_i d_H(\pi_i, \pi_n)}{\sum_{i=1}^{n-1} d_H(\pi_i, \pi_{i+1})}, \tag{9}$$

*where $\Gamma(\mathbf{x}_1, \mathbf{x}_2)$ denotes a path in the activation space between $\mathbf{x}_1$ and $\mathbf{x}_2$. Then, $d_\chi$ is a pseudo-metric.*

*Proof.* Positivity follows from bounds: $d_\chi \in [0, 1]$, as proved for the measure $\chi$ by Lewandowski et al. (2025), symmetry – $d_\chi(\mathbf{x}_1, \mathbf{x}_2) = d_\chi(\mathbf{x}_2, \mathbf{x}_1)$ – follows from the definition (Eq. (9)). To show the triangle inequality, we need to show that $d_\chi(\mathbf{x}_1, \mathbf{x}_2) + d_\chi(\mathbf{x}_2, \mathbf{x}_3) \geq d_\chi(\mathbf{x}_1, \mathbf{x}_3)$ for some $\Gamma(\mathbf{x}_1, \mathbf{x}_2) = \{\pi_1, \ldots, \pi_k\}$ and $\Gamma(\mathbf{x}_2, \mathbf{x}_3) = \{\pi_k, \ldots, \pi_n\}$ (as described in Sec. 3). It is straightforward to check, and requires using $\max_{j \in \{1, \ldots, k\}} d_H(\pi_1, \pi_j) + \max_{j \in \{k, \ldots, n\}} d_H(\pi_k, \pi_j) \geq \max_{j \in \{1, \ldots, n\}} d_H(\pi_1, \pi_j)$. $\square$

$d_\chi$ is a *pseudo*-metric as $\mathbf{x}_1 = \mathbf{x}_2 \Rightarrow d_\chi(\mathbf{x}_1, \mathbf{x}_2) = 0$ but the reverse does not hold, i.e., $d_\chi(\mathbf{x}_1, \mathbf{x}_2) = 0 \not\Rightarrow \mathbf{x}_1 = \mathbf{x}_2$. However, if we restrict $\mathbf{x}_1$ and $\mathbf{x}_2$ such that $\mathbf{x}_1 \not\sim_\chi \mathbf{x}_2$ (Def. 4.4), then $d_\chi$ becomes a metric. Lastly, note that $1 - d_\chi \in [0, 1]$ can be used to measure similarity between input points, and thus may be used in downstream tasks such as clustering or retrieval tasks (see Sec. 7).

### 4.5 Global Space Folding Measure

In this section we adapt a global measure of folding as a median of space folding values along paths that exhibit some folding, i.e.,

$$\Phi_\mathcal{N} := \underset{\{\Gamma : \chi(\Gamma) > 0\}}{\text{median}} \chi(\Gamma) \tag{10}$$

In our experiments on a typical classification task, we have observed that for networks trained to a low generalization error, the folding values $\Phi_\mathcal{N}$ are statistically significantly higher within one the same class than when computed between different classes.[1] This means that, although there is an increasing number of linear regions as indicated by the works which provide bounds on this number, e.g., (Montúfar et al., 2014; Raghu et al., 2017; Serra et al., 2018; Hanin & Rolnick, 2019), the networks fold the space in a very similar manner if the generalization error is low. We formalize it as Conjecture 4.8.

**Conjecture 4.8.** *If a network N achieves near-zero classification error on training data, then it exhibits higher folding on average between classes (higher $\Phi_\mathcal{N}$) than a network with higher error.*

The intuition behind Conjecture 4.8 is as follows. A well-trained network learns to transform intra-class variations of an object such that their representations are brought closer together in the activation space. This effectively *folds* the input manifold for each class into a simpler, more compact structure – an idea reminiscent of contrastive learning, as will be discussed in Sec. 7.

Moreover, we have observed that the folding values are not influenced by the total number of neurons, which we formalize as Conjecture 4.9.

**Conjecture 4.9.** *For sufficiently large networks, $\Phi_N$ approaches a constant that depends only on depth provided low generalization error.*

## 5 Experiments

### 5.1 Evolution of Folding During the Training Process

We start by investigating the behaviour of $\Phi_\mathcal{N}$ (see Eq. (10)) as a function of epoch. We train a ReLU-based MLP ($2 \times 256$) over 100 epochs to a validation accuracy of around 0.5 – the end accuracy is irrelevant as we are only interested in how its *increase* affects values of the folding measure. We store the weights and biases of the model after every epoch of training. We find that the global folding value $\Phi_\mathcal{N}$ is steadily increasing. Interestingly, we observe that, for networks with lower validation accuracy values, the intra-class folding values are similar (or lower) than inter-class folding values, while the opposite holds for well-trained networks. We remark that this is possibly a consequence of the higher concentration of linear regions close to data for well-trained neural networks, as observed by Zhang & Wu (2020). See Fig. 3 for more details.

---

**Algorithm 1:** Interaction Coefficient and Adversarial Attacks.

---

**Input:** Input data of a given label, $(\mathbf{x_i}, y_1)_{i=1}^n$;
**Output:** Values of the interaction coefficient with the number of points
**Step 1:** Perturb the input data $\mathbf{x}$ using a predefined adversarial attack (e.g., PGD, or FGSM);
**Step 2:** Assert that the trained network classifies the data wrongly;
**Step 3:** Compute $\mathcal{I}$ between the original and adversarially perturbed samples;
**return** $\mathcal{I}$

---

### 5.2 Interaction Coefficient

In the next step, we investigate the values of the interaction coefficient $\mathcal{I}$ (Eq. (7)) on the unperturbed and adversarially perturbed images of digits from the MNIST dataset (see Alg. 1). Our intuition here is as follows. Sensitivity of ReLU-based MLPs has been linked to the geometry of their decision boundary (see e.g., Wong & Kolter (2018)); intuitively, the more *ragged* the decision boundary, the more susceptible are the data samples to being mis-classified by the model upon a small perturbation. Such a raggedness of a decision boundary translates to its non-convexity, and the space folding measure was designed to quantify

---

[1] We used the Mann-Whitney test (Mann & Whitney, 1947) to compare intra- and inter-class median folding values in networks with low generalization error. A statistically significant difference (per thresholds in Cohen (1992)) showed that inter-class folding values are higher, suggesting that the network folds space within each digit class for more efficient representation, thus justifying their separate analysis.

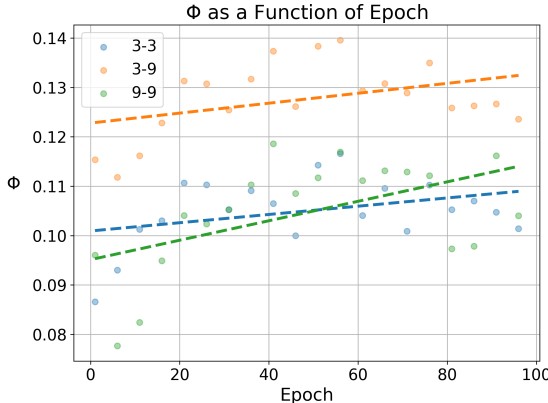

Figure 3: We investigated the values of the global folding measure $\Phi_{\mathcal{N}}$ (Eq. (10)) as a function of epoch. We trained a ReLU-based MLP on CIFAR10 over 100 epochs, and after each epoch we recorded the values of $\Phi_{\mathcal{N}}$. The dashed lines represents fitted linear trends, and serve to illustrate the trend. Note that the folding values keep increasing, even though the end accuracy is around 50%. Note further that we do not observe higher space folding value for inter-class folding (digits $3-3$ and $9-9$); we posit that it stems from a relatively low end accuracy (at initialization the hyperplanes do not concentrate close to data points).

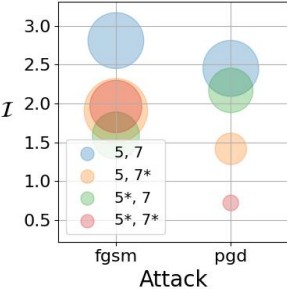

Figure 4: Values of the interaction coefficient (Eq. (7)) were computed using images of digits 5 and 7 from the MNIST test set. The symbol * denotes an adversarially perturbed digit. For visualization, the y-axis has been scaled by a factor of $10^2$. Each circle is centered at the mean of the points it contains, with its size reflecting the number of classified observations.

the *deviations* from convexity (see Lewandowski et al. (2025)), thus studying its non-additivity appears to be a sensible choice to study adversarial geometry.

Across both adversarial attacks considered, the interaction coefficient $\mathcal{I}$ is consistently higher for original (unperturbed) samples (see Fig. 4). Under the projected gradient descent (PGD) attack, the lowest values of $\mathcal{I}$ occur when computed between two adversarially perturbed samples. In contrast, for the fast gradient sign method (FGSM), the interaction coefficients are similar to those obtained when only one of the two samples is perturbed.

## 5.3 Sparsity

We empirically investigate the sparsity according to Def. 4.6 as a function of the total number of neurons. We find that, while sparsity for very small networks (60 neurons total) is quite high (oscillating between 0.4 and 0.8), it rapidly drops for even slightly bigger models (it is stable and close to 0 for models with 480 neurons total). See Fig. 5 for a graphical overview. The results are consistent among different number of layers – the higher total number of neurons $N$, the more paths $\Gamma$ feature the folding effects. We averaged the results over 4 different architectures for each number of total neurons (using dropout or batch-normalization)

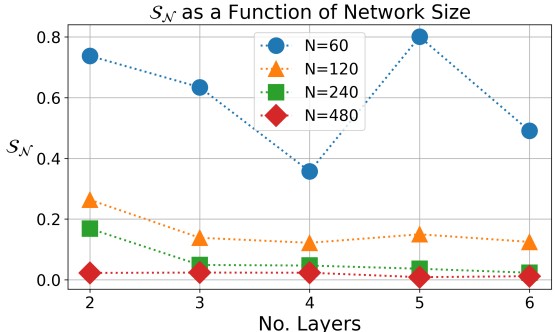

Figure 5: We investigate the sparsity of $\mathcal{S}_\mathcal{N}$ (Def. 4.6) as a function of the number of layers with varying total number of neurons $N$ (denoted by colors).

and found that the variance is very small. Interestingly, the sparsity $\mathcal{S}_\mathcal{N}$ remains remarkably stable across the number of layers for already 480 neurons total, while it varies strongly for networks with 60 neurons.

## 6 Folding as a Regularization

### 6.1 Maximal Folding

Thus far, Lewandowski et al. (2025) observed a strong empirical correlation of space folding values with the generalization capabilities of ReLU-based MLP, motivating its use as a regularization strategy during the training process. Inspired by these findings, in this section we describe our new regularization technique. For the remainder of this section, we will assume that in a path $\Gamma = \{\pi_1, \ldots, \pi_n\}$ every $\pi_i$ is unique, i.e., $\pi_i = \pi_j \Leftrightarrow i = j$. Note that this implies that $d_H(\pi_i, \pi_{i+1}) > 0$ for all $i$. As we intend to maximize space folding we first show that $\chi(\Gamma) \to 1$ for a path $\Gamma$ which oscillates, formalized as Prop. 6.1.

**Proposition 6.1.** *The space folding measure $\chi \xrightarrow{n \to \infty} 1$, where $n$ is the number of steps in $\Gamma$, if for every $\pi_{2k}, \pi_{2k+1} \in \Gamma$ it holds that the Hamming distance relative to the initial point exhibits oscillatory behaviour:*

$$d_H(\pi_1, \pi_{2k+1}) < d_H(\pi_1, \pi_{2k}) \leq C, \quad \forall k \in \mathbb{N}, \ C \in \mathbb{R}_+.$$

*Proof.* Observe that for a path $\Gamma$ oscillating between two activation pattern, $r_1(\Gamma)$ is upper bounded by $\max_k d_H(\pi_1, \pi_{2k})$ (i.e., the maximum Hamming distance between the initial activation pattern and all evenly indexed activation patterns), and $r_2(\Gamma) \to \infty$, thus $\chi(\Gamma) \to 1$. $\qquad \square$

**Proposition 6.2.** *For the fastest convergence of $\chi$ to 1 along a path $\Gamma$ (as defined by Eq. (12)), we need the following condition to be satisfied:*

$$\lim_{n \to \infty} d_H(\pi_1, \pi_{2j}) = c_1, \ d_H(\pi_1, \pi_{2j+1}) = 0. \tag{11}$$

*Denote such a path with $\widetilde{\Gamma}$. Then, for any other path $\Gamma$ it holds that*

$$\lim_{n \to \infty} \frac{\chi(\widetilde{\Gamma})}{\chi(\Gamma)} = 0. \tag{12}$$

The folding measure $\chi$ is not differentiable and thus cannot be used directly during back-propagation based training (however, there are methods that do not rely on back-propagation, e.g. Li et al. (2025)). For traditional training (with back-propagation), the non-differentiable nature of $\chi$ necessitates the use of a surrogate as a differentiable approximation, which we introduce in the next section.

### 6.2 Outline of Implementation

We first remark that $\chi$ is proportional to $r_1$ which suffers from two non-differential components: the max function, and the Hamming distance, and thus is not suitable for a direct use during the training process. However, we note that increasing space folding is equivalent to maximizing the Hamming distances between adjacent $\pi_i, \pi_{i+1}$, which in turn is equivalent to minimizing the cosine similarity. We thus can simplify the learning objective as follows:

$$Loss \leftarrow D_1 \cdot Loss + D_2 \cdot \sum_{i=2}^{n-1} (-1)^{i+1} (\cos(\pi_1, \pi_i) + 1).$$

The term $\sum_{i=2}^{n-1} (-1)^{i+1} \cos(\pi_1, \pi_i)$ encourages monotonicity changes along the path $\Gamma$ with respect to the initial activation pattern $\pi_1$. $D_1$ and $D_2$ are weighting coefficients computed during the training. The resulting folding value approximation can be incorporated into the loss function and optimized using backpropagation. The regularization procedure generalizes to any activation function through equivalence classes (not limited to ReLU-based MLP, see Sec. 4.2).

## 7 Conclusion and Future Work

In our work, we have generalized the concept of space folding to any monotonic activation function, and we have empirically investigated ($i$) the sparsity of paths as a function of total number of neurons, ($ii$) the evolution of global folding measure Eq. (10) during the training process, ($iii$) the behaviour of the interaction coefficient $\mathcal{I}$ (Eq. (7)) on adversarially perturbed samples of MNIST. Our study deepened the mathematical understanding of the space folding measure and lays the groundwork for further experimental work. We have also highlighted key theoretical properties, and suggested its potential as a regularization technique.

We note some parallels between contrastive learning and learning with the space folding regularization. In short, contrastive learning, accomplished through a contrastive loss, ensures ($i$) alignment (closeness) of features between positive pairs, and ($ii$) uniformity of the induced distribution of the (normalized) features on a manifold (Wang & Isola, 2020). Note that, by encouraging higher folding during the training process, we bring data points "closer" in the alignment space, and by penalizing folding between samples from different classes, we push them "further away" from each other, effectively performing contrastive learning. Unlike conventional contrastive loss, which operates on final feature vectors and often requires special architectures or data augmentation, we posit that our approach induces similar benefits (alignment/uniformity) by directly leveraging the network's own folding behavior. It can be seen as a form of contrastive regularization that is naturally available in any labeled training scenario. We will address this hypothesis in an upcoming submission.

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

## A   Sensitivity Analysis

In this section, we analyze the sensitivity of the folding measure $\chi$ along a path $\Gamma$ to perturbations in activation patterns along $\Gamma$. This analysis paves the way to quantifying the impact of clustering the data and then using clusters' centroids to mitigate arising computational complexity (Lewandowski et al., 2025).

**Lemma A.1.** *If each pattern $\pi_i \in \{0,1\}^N$ along path $\Gamma$ is perturbed by $k_i \leq N$ bits (cumulatively $\sum_i k_i$ flips over the whole path), then the folding measure changes at most by*

$$|\chi(\Gamma) - \chi(\Gamma'')| \leq \frac{r_1 + \max_i k_i}{r_2 - \sum_i k_i} - \frac{r_1}{r_2}, \tag{13}$$

*where $\chi(\Gamma'')$ denotes a new, perturbed path, and where we used shorthand notation $r_1 := r_1(\Gamma), r_2 := r_2(\Gamma)$.*

*Proof.* Assume that along a path $\Gamma$, a pattern $\pi_l$ has $k \leq N$ of its $N$ bits changed, and denote the new pattern by $\pi_l'$. By flipping exactly $k$ bits, the maximum change in the Hamming distance between the new activation pattern $\pi_l'$ and any other activation pattern $\pi_i$ on the path $\Gamma$ is $k$, i.e.,

$$\min_i d_H(\pi_l, \pi_i) - k \leq \max_i d_H(\pi_l', \pi_i) \leq \max_i d_H(\pi_l, \pi_i) + k = r_1(\Gamma) + k,$$

and hence the folding measure on the new path $\Gamma'$ satisfies $1 - \chi(\Gamma') \leq (r_1 + k)/(r_2 - k)$, Assume now that along a path $\Gamma$ every data point $\mathbf{x}_1, \ldots, \mathbf{x}_n$ has been perturbed so that their activation patterns $\pi_1, \ldots, \pi_n$ are flipped by $k_1, \ldots, k_n$, $k_i \leq N$ bits, resulting in a new path $\Gamma''$. By analog reasoning as previously, the folding measure on the new path satisfies

$$1 - \chi(\Gamma'') \leq \frac{r_1 + \max_i k_i}{r_2 - \sum_i k_i}. \tag{14}$$

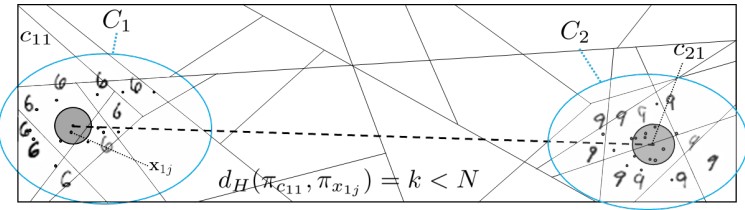

Figure 6: Intra-class clustering to reduce space folding computation complexity in MNIST. Gray regions show the cluster radii for input data. Instead of computing the measure for every data pair (dark dots), we use cluster centroids $c_{ij}$, as described in Alg. 2. In Sec. A we detail the accuracy impact.

Suppose that $\Gamma$ is the path obtained using the original sample, and $\Gamma''$ is the path obtained using the cluster centroids. Rewriting Eq. (14) yields

$$|\chi(\Gamma) - \chi(\Gamma'')| \leq \frac{r_1 + \max_i k_i}{r_2 - \sum_i k_i} - \frac{r_1}{r_2}. \tag{15}$$

$\square$

Lemma A.1 implies that the folding measure $\chi$ is fairly robust to small Hamming perturbations (especially when $r_2 \gg \sum k_i$). This justifies clustering input data and using the respective cluster centroids to mitigate the computational complexity of computing the measure pair-wise between every pair of input data.

---

**Algorithm 2:** Inter-class Clustering Procedure.

---

**Input:** Dataset $\mathcal{D} = \{(\mathbf{x_i}, y_i)_{i=1}^n\}$; Number of classes $L < n$ into which we cluster $\mathbf{x}_i \in \mathcal{D}$;
**Output:** Cluster centroids $c_1, \ldots, c_L$ used to compute the folding measure $\chi$
**Step 1:** Cluster $\mathcal{D}$ into $L$ classes using, e.g., $k$-NN ;       // Complexity: $O(C)$ per inference
**return** Cluster centroids $c_1, \ldots, c_L$

---

We now outline the exact impact of the inter-class clustering on the folding measure. Assume ($i$) the lack of sparsity as defined in Def. 4.6 and ($ii$) symmetrical distribution of maximum value of folding $\chi(\Gamma)$ (observed empirically). Then, writing Eq. (13) for every path between original samples $1, \ldots, j$, i.e.,

$$|\chi(\Gamma_1) - \chi(\Gamma_1'')| \leq \frac{r_1(\Gamma_1) + \max_i k_{1i}}{r_2(\Gamma_1) - \sum_i k_{1i}} - \frac{r_1(\Gamma_1)}{r_2(\Gamma_1)}$$

$$|\chi(\Gamma_2) - \chi(\Gamma_2'')| \leq \frac{r_1(\Gamma_2) + \max_i k_{2i}}{r_2(\Gamma_2) - \sum_i k_{2i}} - \frac{r_1(\Gamma_2)}{r_2(\Gamma_2)}$$

$$\vdots$$

$$|\chi(\Gamma_j) - \chi(\Gamma_j'')| \leq \frac{r_1(\Gamma_j) + \max_i k_{ji}}{r_2(\Gamma_j) - \sum_i k_{ji}} - \frac{r_1(\Gamma_j)}{r_2(\Gamma_j)},$$

adding by sides results in

$$\sum_j |\chi(\Gamma_j) - \chi(\Gamma_j'')| \leq \sum_j \left( \frac{r_1(\Gamma_j) + \max_i k_{ji}}{r_2(\Gamma_j) - \sum_i k_{ji}} - \frac{r_1(\Gamma_j)}{r_2(\Gamma_j)} \right),$$

which establishes an upper bound on the impact of using the clusters' centroids compared to the original pairs of data points. Using cluster centroids introduces at most the error given by Eq. (13) in $\chi$. Summing over all paths in a class for $j = 1, \ldots, J$ clusters) yields an upper bound on the total deviation introduced by clustering. Thus, the loss in accuracy from clustering is bounded, and can be made arbitrarily small by increasing cluster granularity.

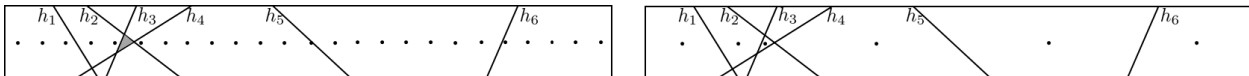

Figure 7: 2D slice of the ReLU tessellation defined by hyperplanes $h_1, \ldots, h_6$. **Left:** Equally spaced points may revisit regions and miss small ones (gray). **Right:** An optimized path visits each region exactly once.

## B  Sampling Strategy

In Lemma 4.3, we have shown the importance of an appropriate sampling for computing the folding measure efficiently (see also Fig. 7). In this section, we introduce a 1D sampling strategy in the Hamming activation space, presented as Alg. 3. Our method is parameter-free, straightforward to implement and intuitive to understand. It is based on the binary search in the Euclidean input space and repetitive checks of the Hamming distance between each of the corresponding activation patterns. We now describe the intuition behind the method. We iteratively bisect the line segment $[x_{\text{init}}, x_{\text{end}}]$ to find intermediate points where the activation pattern flips. If the direct midpoint causes multiple neurons to change (Hamming distance $\geq 2$ between $\pi_{\text{init}}$ and $\pi_{\text{mid}}$), we refine closer to $x_{\text{init}}$ until at most one neuron flips at a time. Then we set that new point as the next milestone and repeat. We successively halve the interval $[x_{\text{init}}, x_{\text{mid}}]$ until the pattern difference drops to 1 bit or we have halved $d$ times (ensuring that we could resolve each differing bit one by one in the best case).

We start by getting activation patterns $\pi_{\text{init}}$ and $\pi_{\text{end}}$ of the two edge points of the path, $\mathbf{x}_{\text{init}}$ and $\mathbf{x}_{\text{end}}$, respectively. Then, while $d_H(\pi_{\text{init}}, \pi_{\text{end}}) \geq 2$, we continuously check $d_H(\pi_{\text{init}}, \pi_{\text{mid}})$, where $\mathbf{x}_{\text{mid}} = \frac{1}{2}(\mathbf{x}_{\text{init}} + \mathbf{x}_{\text{end}})$; if $d_H(\pi_{\text{init}}, \pi_{\text{mid}}) \geq 2$, we assign $\mathbf{x}_{\text{mid}} = \frac{1}{2}(\mathbf{x}_{\text{init}} + \mathbf{x}_{\text{mid}})$, otherwise we accept $\pi_{\text{mid}}$ as the next pattern on the path, and assign $\mathbf{x}_{\text{init}} \leftarrow \mathbf{x}_{\text{mid}}$. For the detailed procedure, see Alg. 3. We note the following nuance to consider. It might happen that the adjacent activation regions have the Hamming distance exceeding 1 – in this case, the corresponding activation pattern might be repeated in the output of our algorithm.

**Complexity Analysis.**  Let $n'$ be the total number of steps actually taken (including refined steps), $O(\texttt{C})$ be the cost of running the network in the inference mode. Hence, the total computational cost is $O(n' \cdot \texttt{C})$. In the worst case, if many boundaries are crossed in very small intervals, the step size keeps halving, leading to a potentially large $n'$. In a typical scenario, $n'$ might be on the order of a few hundred. In a pathological scenario, it can grow larger but is still upper-bounded by repeated halving.

Our method is sequential (each query depends on the last), so it can't generate all samples in one batch like an equal-spaced grid can. However, because it zooms in only where needed, it typically uses far fewer samples than a dense grid would.

## C  Impact of Regularization Techniques

In Fig. 8 we present the results on the folding values when using popular regularization techniques (dropout, batch norm). For the completeness of our study, we have investigated the impact of popular regularization techniques, dropout (Srivastava et al., 2014) and batch normalization (Ioffe & Szegedy, 2015) on small neural networks with a total number of hidden neurons trained on MNIST. All the networks were trained to a high validation accuracy; the values of $\Phi_{\mathcal{N}}$ increase with the depth of the network as reported by Lewandowski et al. (2025). We found that, although networks without batch norm feature higher values of space folding, the differences are small. We thus conclude that the considered regularization techniques do not influence the space folding values in a significant manner.

## D  Additional Results for ELU

In this section we leverage the equivalence relationship defined in Def. 4.2 and compute folding measure on ELU-based MLPs; see Table 2. We train five small neural networks on MNIST to a high validation accuracy

---

**Algorithm 3:** Binary Sampling Procedure.

---
**Input:** Starting point $\mathbf{x}_{\text{init}}$; Ending point $\mathbf{x}_{\text{end}}$; A neural network $\mathcal{N}$ with inference cost $O(\texttt{C})$;
       Maximum number of iterations $T$
**Output:** Intermediate activation patterns under $\mathcal{N}$ between $\mathbf{x}_{\text{init}}$ and $\mathbf{x}_{\text{end}}$
**Step 1:** Obtain initial and target patterns:
$\pi_{\text{init}} \leftarrow \text{GETACTIVATIONPATTERN}(\mathcal{N}, \mathbf{x}_{\text{init}})$
$\pi_{\text{end}} \leftarrow \text{GETACTIVATIONPATTERN}(\mathcal{N}, \mathbf{x}_{\text{end}})$;          // Complexity: $O(\texttt{C})$ per inference
 $\mathcal{P} = \{\pi_{\text{init}}\}$ ;                 // Initialize a list of patterns
**while** $\pi_{\text{init}} \neq \pi_{\text{end}}$ **and** iterations $\leq T$: **begin**
   **if** $d_H(\pi_{\text{init}}, \pi_{\text{end}}) < 2$ **then**
      | **break** ;           // They are sufficiently close; no further refinement
   **else**
      $\mathbf{x}_{\text{mid}} \leftarrow \frac{\mathbf{x}_{\text{init}}+\mathbf{x}_{\text{end}}}{2}$;           // Midpoint in input space
      $\pi_{\text{mid}} \leftarrow \text{GETACTIVATIONPATTERN}(\mathcal{N}, \mathbf{x}_{\text{mid}})$;       // Inference cost: $O(\texttt{C})$
      **if** $d_H(\pi_{\text{init}}, \pi_{\text{mid}}) < 2$ **then**
         // Accept $\mathbf{x}_{\text{mid}}$ as the new starting point
         $\mathbf{x}_{\text{init}} \leftarrow \mathbf{x}_{\text{mid}}$;
         $\pi_{\text{init}} \leftarrow \pi_{\text{mid}}$;
         $\mathcal{P} \cup \{\pi_{\text{init}}\}$;
      **else**
         // Refine $\mathbf{x}_{\text{mid}}$ by repeatedly halving the distance to $\mathbf{x}_{\text{init}}$ until
            $d_H(\pi_{\text{init}}, \pi_{\text{mid}}) < 2$ or attempts exhausted
         $d \leftarrow d_H(\pi_{\text{init}}, \pi_{\text{mid}})$;
         **for** $i = 1$ **to** $d$ **do**
            $\mathbf{x}_{\text{mid}} \leftarrow \frac{\mathbf{x}_{\text{init}}+\mathbf{x}_{\text{mid}}}{2}$;
            $\pi_{\text{mid}} \leftarrow \text{GETACTIVATIONPATTERN}(\mathcal{N}, \mathbf{x}_{\text{mid}})$;       // Another $O(\texttt{C})$
            **if** $d_H(\pi_{\text{init}}, \pi_{\text{mid}}) < 2$ **then**
               | $\mathcal{P} \cup \{\pi_{\text{mid}}\}$;
               | **break**;
         $\mathbf{x}_{\text{init}} \leftarrow \mathbf{x}_{\text{mid}}$;
         $\pi_{\text{init}} \leftarrow \text{GETACTIVATIONPATTERN}(\mathcal{N}, \mathbf{x}_{\text{init}})$;
         $\mathcal{P} \cup \{\pi_{\text{init}}\}$;
**return** $\mathcal{P} \cup \{\pi_{\text{end}}\}$;       // Activation patterns between and including $\pi_{\text{init}}$ to $\pi_{\text{end}}$

---

Table 1: Comparison of sampling methods for discovery of activation regions along a 1D path.

| Method | Vectorizable? | Smallest Discoverable Region |
|---|:---:|:---:|
| Iterative (Gamba et al., 2022) | ✗ | predetermined $\lambda$ |
| Equidistant (Lewandowski et al., 2025) | ✓ | $\|\mathbf{x}_{\text{init}} - \mathbf{x}_{\text{end}}\|/n_{\text{steps}}$ |
| **Ours** | ✗ | $\|\mathbf{x}_{\text{init}} - \mathbf{x}_{\text{end}}\|2^{-T}$ |

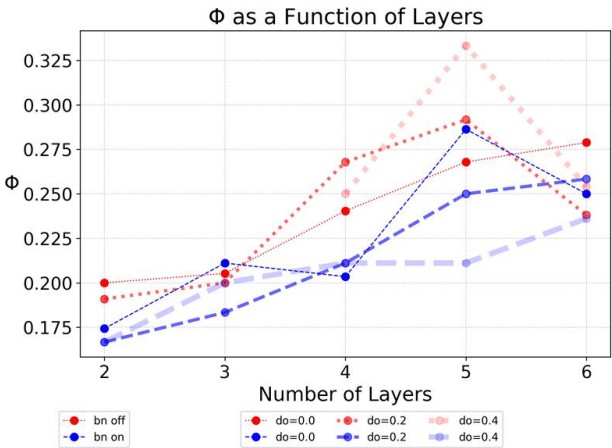

Figure 8: We investigated the impact of batch normalization and dropout with varying rate on Φ. We found that, though there is a small difference, the behaviour remains similar. The thinnest red line corresponds to a network with neither batch norm nor dropout and thus serves as a baseline.

on MNIST, and report the global folding Φ with MAD (Median Absolute Deviation). We find that Φ displays similar behaviour to Φ computed on ReLU-based MLPs.

Table 2: Test accuracy and folding measures across different ELU-based MLP architectures on MNIST. Each model is denoted by $n_{\text{layers}} \times n_{\text{neurons}}$.

| Architecture | Test Accuracy (%) | Mean Φ | Mean MAD |
|---|---|---|---|
| 2×30 | 95.08 | 0.3852 | 0.0964 |
| 3×20 | 95.19 | 0.3091 | 0.0806 |
| 4×15 | 95.04 | 0.3567 | 0.1034 |
| 5×12 | 93.62 | 0.4519 | 0.0611 |
| 6×10 | 93.08 | 0.4648 | 0.0370 |

