# OpenReview forum: "Exploiting Space Folding by Neural Networks"
_TMLR — Withdrawn by Authors_

### Review · Reviewer_FAjf · 2025-06-07

**Summary Of Contributions:**

This paper studies a path-folding measure proposed in previous work by Lewandowski et al. 2025, and relaxes it to propose a new regularization strategy for deep ReLU networks.

**Audience:**

Yes

**Claims And Evidence:**

No

**Requested Changes:**

- I am curious if the authors can relate their analysis to other studies of representational distances along interpolated paths between inputs. In particular, [Novak et al 2018](https://arxiv.org/abs/1802.08760) considered Jacobian norms and transitions between linear regions in ReLU networks, while [Zavatone-Veth et al 2023](https://arxiv.org/abs/2301.11375) and [Yang et al 2024](https://arxiv.org/abs/2405.17181) considered the pullback to pixel space of the Euclidean metric on representation space.

- Definition 4.4 is not clearly phrased, as eq. (6) includes both the notation for the equivalence relation and its definition, which does not fit into the sentence above.

- The relevance of the work of t Balestriero & Baraniuk (2018) to the present analysis is not made clear by the vague discussion on Page 5. How does their work help justify or contextualize the proposed extension to non-ReLU networks via binarization?

- This is not crucial for acceptance, but I wonder whether Conjecture 4.5 can be proved at initialization, i.e., with random weights.

- Can you make Conjecture 4.9 more precise? In particular, what do you mean by "provided low generalization error"? What is the generalization error low relative to?

- This is not crucial for acceptance, but including exploratory experiments using a broader array of network architectures and datasets (beyond MLPs trained on MNIST) would enhance the paper.

- I do not follow how one would differentiate through the binarization required to define the activation pattern in the regularizer as defined in Section 6.2. Do you propose to use a surrogate gradient for the derivative of the step function?

- A set of experiments which I do view as crucial for acceptance are tests of the proposed regularization strategy. The authors should ideally include results for multiple architectures, and must definitely show results for datasets other than MNIST. Depending on whether they want to make claims regarding adversarial robustness, they should also compare against previously-proposed methods for regularization in adversarial training (which are too numerous to list here; the authors may choose a set that they deem appropriate).

**Strengths And Weaknesses:**

The paper is generally clearly written (other than the few issues noted under Requested Changes), and is certainly of interest to the TMLR audience. I will keep my review brief, as I have a single major concern that precludes my recommending acceptance: the authors do not evaluate the efficacy of their proposed regularizer. This could be easily addressed with even a few moderate-scale experiments (see Requested Changes). Experimental tests of the regularizer are vital both to justify why others should test it, and to bring the narrative to a satisfying close.

---

### Review · Reviewer_BLU4 · 2025-06-14

**Summary Of Contributions:**

The authors analyze how the input space of ReLU MLPs is represented in folded form in the hidden unit activation space. Using a cheap-to-compute measure of such deformation which considers the growth and shrinkage of distance in activation space when traversing along a 1-d line in input space, originally proposed by Lewandowski et al. (2025) and further developed here, they conduct several proof-of-concept experiments: Evolution of the measure over training, folding of paths connecting samples within and between classes, of paths to adversarially perturbed samples, and a regularization term encouraging folding of pre-defined paths.

**Audience:**

Yes

**Broader Impact Concerns:**

/

**Claims And Evidence:**

No

**Requested Changes:**

- The generalized folding measure (listed in the main contributions on p.2) treats every nonlinearity as ReLU-like, assuming that the main bend of the function occurs at $0$. However this is not the case for the common sigmoidal activation function. Furthermore, the generalized version is only proposed but never used and therefore currently adds little to the manuscript.
- The regularization (listed in the main contributions on p.2) is written as an idea and only an "implementation outline" is given, but it is never used. The link of the regularization to contrastive learning (advertised in the abstract) is vague and appears only in the outlook section.
- I had trouble understanding how the defs 4.1, 4.6, 4.8 in sect.4 are meant, and in particular whether the path in activation space is fixed to always correspond to the one generated by following from linear region to linear region along a straight line between $x_1, x_2$ in input space, or not. While the straight line is mentioned in the beginning of sect.4.1 and used later in the experiments, this straight line is never referred to in the definitions. Instead, these always refer only to "a path" in the activation (Hamming) space, as if it was not fixed given a network and a pair of points in the input. If on the other hand "a path" refers to any non-straight path between $x_1, x_2$, then the measures are not uniquely defined, e.g. the pseudo-metric prop.4.7.
- Generally, the presentation could be much clearer on why the construction of the measure makes most sense in the way it is done in the manuscript, based on the way a ReLU network tiles the input space with activation patterns. Figs. 1 and 2. where not so clear in illustrating this, and also increased my confusion about the uniqueness of the path considered (the caption of Fig.1 mentioning that the shortest path is not unique, which is true of course, but why is the path with arrows shown in Fig.1.left, if one is interested only in the dotted path?). In Fig.1.right, it is unclear to me why the grey regions are closer to each other than to the $\pi_j$ region in between, if I assume that the step in Hamming distance moving from one region to an adjacent one is always one. In a ReLU network, in principle when moving from one region to the adjacent region, the step in Hamming distance is always 1, if one does not step exactly through an intersection of hyperplanes. But it is possible to cross in and out of a hidden unit's activation region because the corresponding hyperplane in the hidden layer pre-activation space of the unit does not have a planar image in the input space of the network, as the plane's image does not continue straight through the intersection with other plane's images. Only in this sense, I think, can the path on the hypercube induced by following a straight line in the input have a non-monotonic Hamming distance evolution. If my understanding is correct, would it maybe be better to visually explain this in Fig.1.right? In Fig.2, The grey dotted Hamming distance evolution makes steps which seem not to be $+/- 1$ (or at least $d_H$ is not clearly an integer), and I was missing a correspondence of the black and grey-dotted vectors on the horizontal axis.
- (optional) Investigating the relation between adversarial examples and the folding measure or the interaction coefficient in more depth could make the paper much stronger. Fig. 4 alone is not sufficient to draw any conclusions. Are these findings representative, at least for a certain task and architecture? Additional data and analysis would be needed. The same issue exists with Fig.3.
- Currently the experiments are insufficiently described and not reproducible. The main text does not give a clear description of what was done ( e.g. sect.5.1 does not state what task the MLP is trained on but gives unclear specifics about the validation accuracy, footnote 1 on p.8 claims some significance test was done but it is not clear on what and no results/data are shown, sect. 5.3 also does not state the task or specify what set of paths is considered, etc...). Partly this information exists in the appendices but is not referenced in the main text, and it is completely unclear where to find what.
- The appendices are not referenced, and their motivation or results not discussed in the main text.
- For reproducibility, please provide the code for the experiments.

### Small questions and comments

- Def.4.4 says "[...] equivalent under $\chi$ IF" and then an equivalence relation (6) is written, which I assume is meant not as a condition, but as the definition of the equivalence?
- Conj.4.8 says higher between classes, while the text just above refers to higher folding values within classes.
- The definition of the interaction coefficient is concerned with interaction between two paths, but Alg.1 and Fig.4 seem to be concerned with interaction between two points? (At least it was unclear to me what paths where used here)
- In the sampling procedure of the points along a straight path, Alg.3 and App.B, is it not possible that the Hamming distance becomes $d_H (\pi_{\mathrm{init}}, \pi_{\mathrm{end}}) = 1$ even though there is still more than one activation boundary in between? Though possibly this is rare in larger networks.

**Strengths And Weaknesses:**

### Strengths

- Development of interesting low-dimensional quantities to probe the high-dimensional geometry of learned representations
- Proof-of-concept ideas for applying these new tools to study adversarial perturbations and how class manifolds evolve through training.


### Weaknesses

- Beyond proposing the various observables, the manuscript is a collection of preliminary ideas (e.g. conjectures 4.5, 4.8, 4.9, the regularization method), rather than presenting a systematic investigation (theoretical or experimental) of any of these.
- The methods and experiments were not each clearly described. Also the content of most appendix sections is not mentioned in the main text and I had to search back and forth to get an idea of how the experiments where actually done, and what information was provided or omitted.

In sum, the manuscript provides interesting concepts and explorative ideas but not a comprehensive theoretical or empirical analysis, and could be more clearly written.
I think the manuscript could be a much stronger contribution if additional work is put into the presentation and into empirically probing one or two of the claims in more depth. If the authors would prefer not add further experiments, at least the clarity of presentation must be improved. Please find my detailed questions and comments below.

Please note that I did not validate the calculation in App. A.

---

### Review · Reviewer_wgpb · 2025-06-17

**Summary Of Contributions:**

This paper investigates how neural networks transform the input space through their internal representations. Building on prior work that suggests ReLU networks fold the input space—bringing distant inputs closer in activation space—and that higher-accuracy models tend to exhibit more folding, the authors extend the folding metric to a broader class of monotonic activation functions. They provide a theoretical analysis of the folding measure and introduce additional metrics to capture properties of the network’s decision boundaries and the degree of spatial folding.

On the empirical side, the paper studies how these measures evolve during training in ReLU networks and under varying hyperparameter settings. Finally, the authors propose a differentiable regularization method to promote folding.

**Audience:**

Yes

**Claims And Evidence:**

Yes

**Requested Changes:**

See above.

**Strengths And Weaknesses:**

The paper introduces a well-motivated framework for studying the geometry of internal neural networks representations and the tools discussed in this paper can offer new insights.

However, several parts of the paper seem vague to me and addressing these issues would help better understand the framework:

1. **Thresholding in folding metric for monotonic activations:** the folding measure relies on binarizing the activations using a zero threshold. Thus, the extension applies to monotonic activations that have non zero values in their range. For example, that’s not the case for an activation function like sigmoid.
Should the threshold value always be fixed at zero? If so, does it have a geometric interpretation? How sensitive are the results to this threshold?

2. **Definition and computation of folding-based metrics:** Equations (8) and (10) depend on a set of paths. What exactly are these paths? Are they computed between random sample pairs, training pairs, or test pairs? Or is the theoretical formulation intended to hold over all pairs in the input distribution? Empirically, does the value of the folding measure differ when evaluated on training vs. test samples?

3. **Section 5.1:** The folding trend is tracked only up to 50% accuracy. Is there any guarantee that the same trend continues with further training? Also, in this section it’s mentioned that “we observe that, for networks with lower validation accuracy values, the intra-class folding values are similar (or lower) than inter-class folding values, while the opposite holds for well-trained networks.” However fig 3 only shows one of these trends. Where do you report the opposite?

4. **Connection between Sections 6.1 and 6.2:** i feel slightly lost figuring out the connections of the regularization motivation in 6.1 and the actual proxy introduced in 6.2. Also, is there any empirical evidence that this folding proxy regularization term improves training?

5. **Section 5.2, Fig 4:** How exactly is the interaction coefficient computed? Specifically, this metric is defined between two paths.  How is the connecting point between them chosen?
Also, i understand at an intuitive level that the high values of this metric between the clean samples indicates the decision boundary is ragged. However, I find it less obvious how to interpret the lower values for perturbed samples. What does this measure capture in this case?

---

### Note · Authors · 2025-07-01

**Comment:**

We thank reviewers for their time and valuable feedback on our paper. After careful consideration, we have decided to withdraw our paper.

**Withdrawal Confirmation:**

I have read and agree with the venue's withdrawal policy on behalf of myself and my co-authors.